# Connectivity assessment and prioritization of urban grasslands as a helpful tool for effective management of urban ecosystem services

**Hassanali Mollashahi**[1]*, **Magdalena Szymura**[1], **Tomasz H. Szymura**[2]

**1** Institute of Agroecology and Plant Production, Wrocław University of Environmental and Life Sciences, Wrocław, Poland, **2** Department of Ecology, Biogeochemistry and Environmental Protection, University of Wrocław, Wrocław, Poland

* hassanali.mollashahi@upwr.edu.pl

**Data Availability Statement:** All relevant data are within the manuscript and its Supporting Informationw files.

## Abstract

Urban grasslands are usually managed as short-cut lawns and have limited biodiversity. Urban grasslands with low-intensity management are species rich and can perform numerous ecosystem services, but they are not accepted by citizens everywhere. Further, increasing and/or maintaining a relatively high level of plant species richness in an urban environment is limited by restricted plant dispersal. In this study, we examined the connectivity of urban grasslands and prioritized the grassland patches with regard to their role in connectivity in an urban landscape. We used high-resolution data from a land use system to map grassland patches in Wrocław city, Silesia, southwest Poland, Central Europe, and applied a graph theory approach to assess their connectivity and prioritization. We next constructed a model for several dispersal distance thresholds (2, 20, 44, 100, and 1000 m), reflecting plants with differing dispersal potential. Our results revealed low connectivity of urban grassland patches, especially for plants with low dispersal ability (2–20 m). The priority of patches was correlated with their area for all dispersal distance thresholds. Most of the large patches important to overall connectivity were located in urban peripheries, while in the city center, connectivity was more restricted and grassland area per capita was the lowest. The presence of a river created a corridor, allowing plants to migrate along watercourse, but it also created a barrier dividing the system. The results suggest that increasing the plant species richness in urban grasslands in the city center requires seed addition.

## Introduction

Urban green space provides a variety of important ecosystem services [1, 2] ranging from conservation of biodiversity [3], maintenance of landscape connectivity [4], aesthetics [5], leisure and recreation [6], and human health benefits [7, 8]. It also shapes microclimates by mitigating urban heat islands [9, 10]; improving soil, water, and air quality [11, 12]; and reducing stormwater runoff [13]. Grasslands, primarily represented by urban lawns, constitute an important component of urban green space [14].

**Funding:** Publication financed by the project "UPWR 2.0: international and interdisciplinary programme of development of Wrocław University of Environmental and Life Sciences", co-financed by the European Social Fund under the Operational Program Knowledge Education Development, under contract No. POWR.03.05.00-00-Z062 / 18 of June 4, 2019.

**Competing interests:** The authors have declared that no competing interests exist.

Frequently cut urban lawns are a worldwide phenomenon of the urban landscape and represent a significant part of urban greenery. The lawns, which serve as functional and accessible areas in parks, playgrounds, and private gardens, have a generally positive public reception. However, the intensive grass-cutting regime has a negative impact on the urban environment [15], and scientists and greenery managers are increasingly considering species-rich, low-intensity urban grasslands and grass-free urban lawns. Low-intensity management benefits the richness of plant species in urban grasslands [16–19]. This richness, paired with the vegetation height of urban grasslands, positively influences soil microbial communities [20] and the diversity of arthropods, including pollinators, and provides advantages for other animals such as birds [21–24]. For example, the abundance of parasitic *Hymenoptera* is associated with herb diversity, and these insects are one the most important biocontrol agents providing natural pest management services in urban landscapes [25]. Species-rich urban grasslands with low-intensity management boost the resilience of the ecosystem, which enhances its ability to accumulate carbon and nitrogen [26, 27], and reduces the public cost for maintenance [23, 28–30]. Moreover, the high plant diversity directly increases human well-being and offers psychological benefits [31–35].

Urban grasslands are usually species poor because of intense management and the use of standard, species-poor seed mixtures in the original plantings [15, 17, 18]. In urban areas, ecologically desirable species are often completely absent because natural disperser vectors and source populations are largely missing [28, 36–40]. Additionally, the soil seed bank does not significantly contribute to the desired floristic development, especially on young, heavily altered urban soils. Therefore, the ability of urban sites to function as novel habitats for grassland species may be limited by spatial isolation and missing diaspore pools [41]. Because of these limitations, the restoration of urban grasslands mostly relies on the establishment of entirely new grasslands or by seed addition to existing grasslands [17, 19, 41, 42]. However, maintaining and/or increasing biodiversity can be restricted by limited dispersal caused by the high isolation and low connectivity of grassland patches within the urban landscape [39].

The connectivity between patches is important for maintaining a vital population, as well as allowing interaction between species. Greater connectivity between habitat patches contributes to genetic diversity, especially among insect-pollinated and outcrossing plant species [43]. In contrast, habitat fragmentation can lead to the extinction of species due to inbreeding [44, 45]. Consequently, establishing a properly managed biotope network is important in nature conservation strategies, as well as provision of ecosystem services related to landscape connectivity [46–49]. The concept of landscape connectivity, defined as the degree to which the landscape facilitates or impedes the movement of species between patches, allows us to understand how organisms disperse and to predict where they go [50, 51]. Connectivity encompasses two elements: functional and structural. Plant functional connectivity pertains to the effective dispersal of propagules or pollen between habitat patches in a landscape, while structural connectivity describes the physical aspects of the landscape (e.g., size and proximity of patches) and the configuration of habitat patches [51, 52]. Structural connectivity is not a simple concept [53], but it provides useful information for policy makers and managers to plan management strategies [39, 54–56]. To date, several projects [57, 58] and management actions have been introduced, boosting the provision of ecosystem services that have cost-effective outcomes [59, 60]. A good example of practical solutions for managing urban greenery is Detroit, Michigan, in the United States, where the concept of structural connectivity, a cost-effective plan for reconnecting isolated habitat patches, has been expanded on to maximize social and ecological functions [4, 61].

Species-poor conventional lawns are a widely accepted anthropogenic construct that is traditionally maintained as an aesthetic standard by intensive management [62, 63]. A common

view among the majority of the population and even among decision-makers is that low-intensity grasslands are a sign of neglect and laziness, perhaps because they appear "messy" [16, 64]. In Poland, short-cut lawns are considered a symbol of financial status [65]. Moreover, low-intensity management can reduce the recreational value of urban grasslands [28], and therefore, it cannot be introduced everywhere. Consequently, grasslands need to be prioritized to obtain the most effective network of environment-friendly, species-rich urban grasslands.

In this study, we analyzed the spatial structure of urban grassland patches in the city of Wrocław, Poland, with a focus on the connectivity between the patches. We took into account the functional connectivity by considering various dispersal distance thresholds for both seeds and pollen, irrespective of the particular plant species. The detailed aims of the study were (a) to examine how the connectivity between grassland patches can change based on different dispersal distances, (b) to determine the prioritization among the different patches based on landscape-scale connectivity, and (c) to determine how the spatial patterns of grassland distribution and connectivity are related to human population density in the city.

## Materials and methods

### Study area

Wrocław is located in Silesia, southwest Poland, Central Europe (51˚ 6″ 28.3788″ N, 17˚ 2″ 18.7368″ E). The total area is about 300 km$^2$, and the city's population is approximately 650,000. The city is located on the Odra River and lies mostly in the river valley at an altitude ranging from 105 to 156 m a.s.l. The average annual temperature is 9.7˚ C, and the annual precipitation is 548 mm, with most occurring as rainfall in the summer. July is the warmest month, with an average temperature of 19.9˚ C, and January is the coldest month with an average temperature of about −0.5˚ C. A typical urban heat island is observed in the city. The city is surrounded by a relatively uniform landscape of intensively used agricultural areas and narrow strips of riparian forest and seminatural vegetation along watercourses, which represent the main ecological corridors.

The system of urban green areas consists of urban forests, parks, allotment gardens, and grasslands. The total acreage of urban greenery is 15,648 ha, and it accounts for 53% of the Wrocław city land. The grasslands include public and private lawns, road verges, and grasslands on river embankments. There are also three special areas that are dominated by grasslands: so-called irrigation fields, the airport, and the aquifer area (Fig 1). The irrigation fields, which were used up to the 1990s for wastewater cleaning (septic drain fields), include approximately 10 km$^2$ of semi-natural vegetation, mostly grasslands and reed-beds. The fields are owned by the city and are kept for nature protection. The grasslands related to the Wrocław Airport infrastructure covers about 3.5 km$^2$. The aquifer area, as a protected part of the river catchment area used to supply drinking water for the city, covers an area of 2 km$^2$.

A considerable area of grasslands related to river banks and flood areas is owned by the state and is managed by the Regional Water Management Board. The urban grasslands in Wrocław are usually managed intensively with cutting several times per year, and, in the case of Wrocław Airport, by spraying herbicides. The exceptions are irrigation fields and aquifer areas, which are maintained with low-intensity management, with cutting once or twice per year. Low-intensity management of urban grasslands has been recently introduced by the city authorities, but it has not been widely accepted by the citizens.

### Data sources

We used the Polish Database of Topographic Objects (BDOT10k), which collects data on different kinds of topographic objects, including the land-cover class "grassland" [66]. The

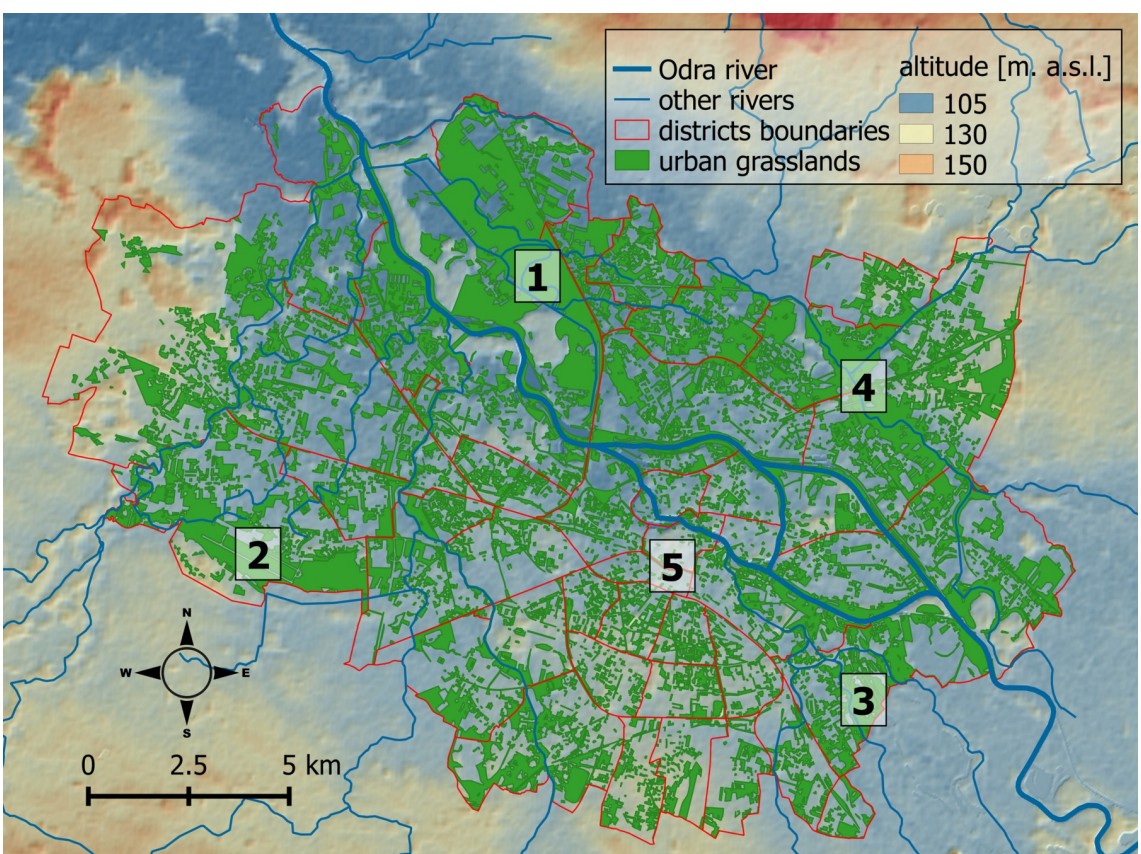

**Fig 1. The urban grasslands of Wrocław city with altitude shown in the background.** The numbers denote the following areas: (1) irrigation fields, (2) airport, (3) aquifer areas, (4) areas managed by the Regional Water Management Board, and (5) the city center. The shapefiles are provided by Polish Database of Topographic Objects [66] and Wrocław Spatial Information System [69]. The land relief layers were produced using data from EU-DEM [71].

BDOT10k database roughly corresponds to a map at 1:10,000 scale and is regularly updated. The minimum size of a mapped patch is 1000 m², and in the case of linear features (e.g., road verges), the width must exceed 5 m. This map is considered the most comprehensive database regarding the distribution of urban green space [67] and grasslands in Poland [68]. The original map in "shp" format was cropped to the administrative boundaries of Wrocław city and checked for invalid polygon geometries. The invalid geometries were fixed, and the map was used for further analysis (Fig 1).

The map of Wrocław district boundaries was provided by Wrocław Spatial Information System [69]. For calculation of grassland area per capita, we used data on population density in Wrocław city districts obtained from the Public Information Bulletin of the Municipal Office of Wrocław [70].

## Methods

**Dispersal distance threshold.** Estimating the dispersal ability of a plant species is difficult because dispersal traits such as pollen vector (wind, insect, self-pollination), reproduction (seed, vegetative growth, and mixed seed and vegetative), and dispersal mode (anemochory, barochory, endo- and/or epi-zoochory) may affect it [61, 72, 73]. Thomson [74] showed that the mean dispersal capacity of plant species was 203 ± 23 m based on biotic vectors, while it

was 44 ± 34 m for species dispersed abiotically. Donath [75] found median dispersal distances of 13–50 m for most grassland species.

Because our study focused on the connectivity between grassland patches with different plant species, we considered a range of dispersal distances. Following the approach of Hejkal et al. [39], we set four dispersal distances as follows: 2, 20, 44, and 100 m. We assumed that the distance thresholds were adequate for our study site, because central European cities have very similar flora and urban environments [76, 77], especially in the case of urban grasslands [17–19]. In addition, we also included the threshold distance of 1000 m because gene flow by pollen is part of the plant functional connectivity [51], and this distance has been suggested for pollen movement for many plant species [53, 78].

**Connectivity analysis and patches prioritization.** Various methods are used to model the connectivity in an urban landscape [54, 79]. Euclidean distance and different connectivity indices are based on the geographic distance [80], gravity theory [81], a least-cost path approach [82], and graph theory [83, 84], with specific forms of the latter including network and circuit theories [85, 86]. Among the methods, graph theory offers powerful and effective tools for representing landscape patterns in a quantitative way as well as performing complex analyses regarding landscape connectivity [86]. In this approach, connectivity is represented by groups of habitat patches (i.e., *nodes*) and the links that connect paired nodes, including the movement between them. Connectivity is assumed to exist and to be unrestricted within each node (so-called intrapatch connectivity). The links encode information about the physical distances among patches and can represent structural, potential, or functional connection. Additional information regarding the dispersal abilities of focal species (e.g., maximum dispersal distance threshold) can be used to eliminate links (e.g., those exceeding the threshold distance) and finalize a representation of the potential connectivity for the focal species [86]. The term *component* considers a set of nodes (i.e., habitat patches) connected by links and thus defines a group of patches with possible migration within the system. An isolated patch is itself a component [87]. Within this general framework, many different connectivity indices have been used, including the simplest and most intuitive, such as the number of existing links between patches (NL) and the number of components (NC) representing the number of groups of patches which are connected. Both indices can be calculated by considering the maximal dispersal threshold for focal species. A higher NL and a lower NC denote better connectivity. One of the more sophisticated indices is the integral index of connectivity (IIC), which was proposed by Pascual-Hortal and Saura [87] and is considered to be very effective. This index offers a quantitative basis for adequately prioritizing the conservation of landscape elements (patches and links) that are particularly critical for maintaining the overall habitat connectivity. Therefore, the IIC allows not only estimating the current "degree of connectivity" within a landscape, but it also offers a relative ranking of patches by their contribution to overall landscape connectivity [87]. This relative ranking is considered to be the most useful tool in the decision process for planners [84, 88, 89]. The importance of each patch in overall connectivity (IIC) was assessed based on the difference (delta, d) in the IIC value when that patch was excluded from the entire system. The rank of dIIC values for each patch ranged from 0 to 1, with a higher value indicating greater importance of the patch for connectivity of the analyzed landscape [87]. Moreover, the dIIC index enables distinguishing three fractions, which additively yield the overall value. The first fraction includes the intrapatch connectivity component (*intra*), which is based on the assumption that connectivity exists within the patch. Two fractions compose the interpatch connectivity component: *flux*, which indicates whether the node is directly connected to other nodes, and *connector*, which indicates whether a node serves as a stepping stone and contributes to the connection between other nodes [90].

For the connectivity analysis, we used Conefor Sensinode 2.6 software. The input data set was created using Conefor Inputs plugin in QGIS software. Three connectivity parameters were calculated: the number of links between patches (NL), the number of components (NC), and the integral index of connectivity (IIC), which reflects the overall connectivity [91]. All metrics represent the system for specific and assumed dispersal distances [91]. For NC, we also calculated the number of components for which the sum of the area exceeded 50% of the total area of urban grasslands, for each distance threshold. We used this approach because some components consisted of only a few small patches, while others encompassed several large patches. Therefore, we considered the number of components that covered 50% of the area as better reflecting the landscape structure from a biological perspective. The prioritization of patches was assessed based on dIIC calculations. For comparing the relative role of a particular fraction in the dIIC index, we recalculated the values of $dIIC_{intra}$, $dIIC_{flux}$, and $dIIC_{connector}$ as percentages, and the average value of a particular fraction for each distance threshold was computed.

## Results

The Wrocław urban grasslands, with an area of 9,523 ha, constituted 60% of the urban green areas and 32% of entire city area. The grassland patches were distributed across the city, but they were more abundant in the northern part of the city and in the peripheries. The largest grassland patches were mostly in the northern part, with a few in the southern and southwestern parts of the city (Fig 1). A total of 2442 grassland patches were analyzed. The size of the smallest grassland patches was 0.003 ha and the largest was 1179 ha. The smallest patches were the most numerous, with a size up to 0.5 ha, but the sum of their areas was low. Almost half of the total grassland area in the city belonged to a few patches that were larger than 100 ha (Fig 2). The median grassland area was 0.4 ha.

The results of the connectivity analyses revealed considerable changes of connectivity indices with assumed dispersal distance thresholds (Table 1). For the smallest dispersal threshold (2 m), the number of links between patches was only 11, causing the urban grasslands to form 2431 isolated components. The number of links increased as the distance threshold was changed from 2 to 1000 m, which indicates better connectedness of species with a greater dispersal ability. Even with larger dispersal distances, however, there were still numerous separated groups of patches (components): 1640 components for a distance threshold of 20 m, 666 for a threshold of 44 m, and 126 components for 100 m. Only with a dispersal threshold 1000

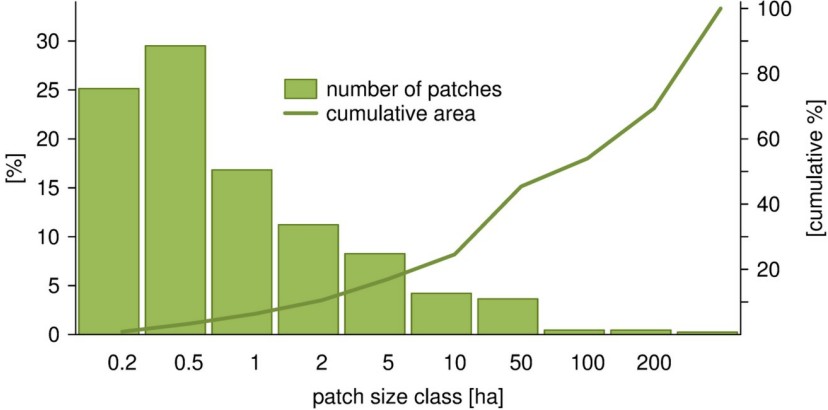

**Fig 2. The distribution of grassland patches in Wrocław in terms of the number and cumulative area in patch area classes.** Note the semi-logarithmic scale in patch area classes.

**Table 1. Results of connectivity indices for different dispersal distance thresholds for urban grasslands in Wrocław city.**

| Indices[a] | Dispersal distance thresholds [m] | | | | |
|---|---|---|---|---|---|
| | **2** | **20** | **44** | **100** | **1000** |
| NL | 11 | 867 | 2259 | 4480 | 61,762 |
| NC [b] | 2 431 (43) | 1 640 (4) | 666 (2) | 126 (1) | 2 (1) |
| IIC | 2,468,023 | 4,780,829 | 7,485,704 | 14,984,170 | 22,666,110 |

[a]NL, number of links; NC, number of components; IIC, integral index of connectivity.

[b]The number of the largest components for which the sum of the area exceeded 50% of total urban grassland area is shown in parentheses.

m was the connectivity very extensive, with only two components. Taking into account not only the number of components but also their area, we found that a considerable fraction (>50%) of all urban grassland area consisted of a smaller number of components: 43 in the case of the 2-m distance threshold; four and two for 20 and 44 m, respectively; and up to one for distances of 100 and 1000 m (Table 1 and Fig 3). For smaller distance thresholds (up to 44 m), the Odra river quite often separated the urban grasslands into disconnected components (Fig 3A and 3B).

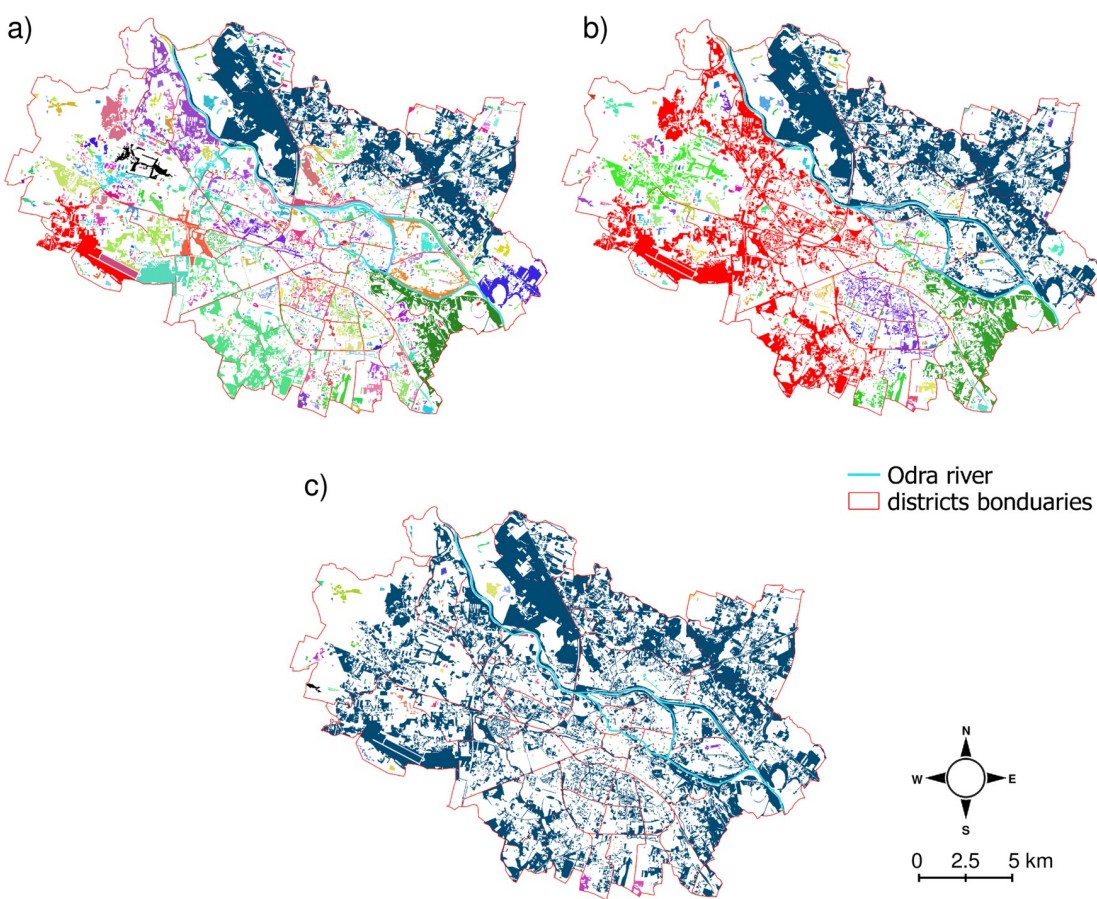

**Fig 3. Spatial distribution of components in Wrocław urban grasslands.** Different colors indicate particular components (groups of patches that are connected) calculated for distance threshold: (a) 20 m, (b) 44 m, and (c) 100 m. For simplicity, the smallest and largest distance thresholds (2 and 1000 m, respectively) are not shown. The shapefiles are provided by Polish Database of Topographic Objects [66] and Wrocław Spatial Information System [69].

Examination of the relative importance of particular components for overall values of dIIC showed correlations with dispersal distance thresholds (Fig 4). For the 2-m distance, the dIIC value was almost entirely influenced by patch size (intra component). With regard to the intra component, the importance of patch size decreased as dispersal distance increased, while the influence of connection between a patch and other patches (flux component) exhibited a significant increase when the dispersal distance became greater. In the system of urban grasslands of Wrocław, the role of patches creating stepping stones for connections between other patches (connector) was rather small and had the highest influence for a dispersal distance of 44 m (Fig 4).

Nonetheless, all components of dIIC, the dIIC values themselves, and the area of patches were correlated for all dispersal distance thresholds. As a result, the large grassland patches usually had the highest dIIC values, regardless of the distance threshold (for detailed results, see S1 Table). As an example, Fig 5 shows the location of 20% of the most important patches for connectivity (i.e., those with the highest dIIC values) in Wrocław city for the 44-m distance threshold (maps for all distance thresholds are shown in map A-E in S1 File. Such spatial configuration of urban grasslands meant that the grassland patches in the Wrocław city center were isolated from other grasslands and had very low dIIC values.

In Wrocław the grassland area per capita ranged from 13 m$^2$ in the central districts, to 6592 m$^2$ in the suburbs. We observed a negative relationship between grassland area per capita and human population density (Fig 6). The highest population density was found in central districts (14,025 people/km$^2$), where sparse, small grassland patches occurred, while the area and number of grassland patches were higher in the suburbs, where the population density was lower, reaching the minimum value 83.1 people/km$^2$ (for details, see Map in S2 File).

## Discussion

### Urban grassland amount and distribution

The percentage of urban green area in Wrocław city is relatively high compared with other European cities, where it usually ranges from 2% to 46%, providing 3 to 300 m$^2$ of green area per capita [92]. Similar to other cities in the world [29, 64, 93], urban grasslands in Wrocław cover a considerable area and constitute a dominant component of urban greenery. The results of our study highlighted that the spatial distribution of grassland patches caused most of the city population to be separated from grasslands. The proportional decline of green space

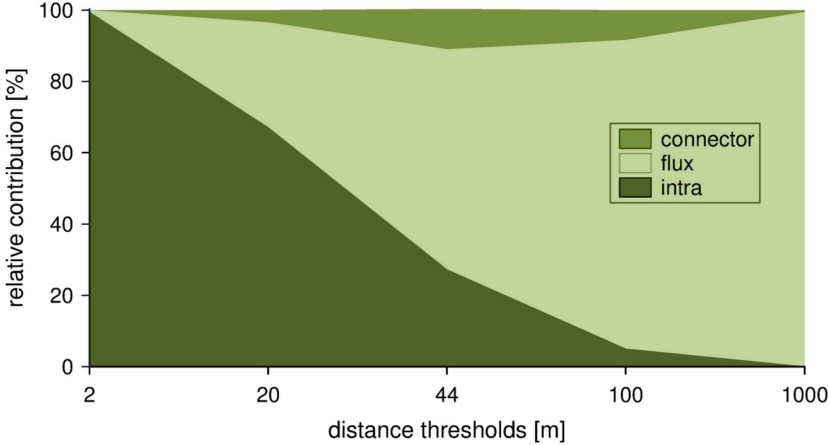

**Fig 4. Relative contribution of each dIIC fraction (connector, flux, intra) on the total importance of an individual patch along with distance thresholds.**

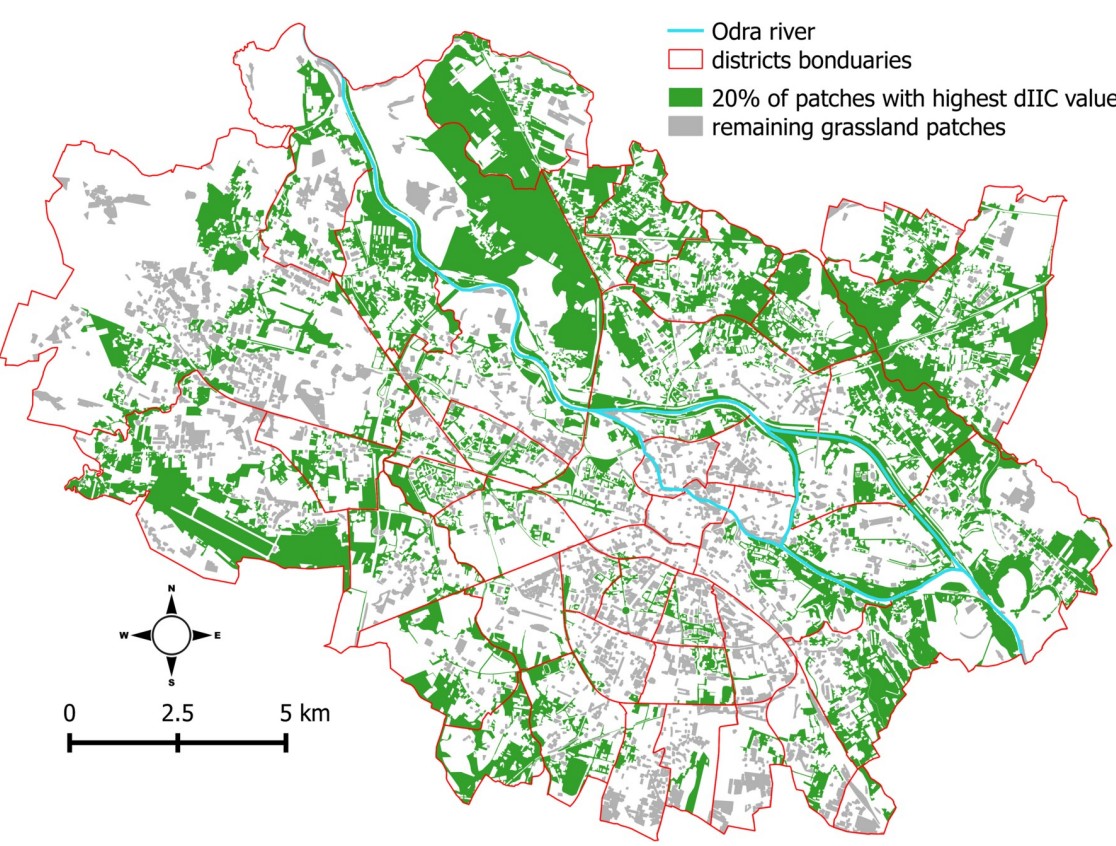

**Fig 5. Location of urban grassland patches in Wrocław city.** The green color shows the 20% of grasslands patches with the highest dIIC values for a dispersal distance of 44 m; the gray indicates the remaining 80% of patches. Notably, the patches with the highest dIIC values are the largest. It is also clear that patches in city center usually had a low value for overall connectivity. The shapefiles are provided by Polish Database of Topographic Objects [66] and Wrocław Spatial Information System [69].

coverage with an increase in human population can be considered as the general pattern of greenery in European cities [92]. It is particularly true for lawns, where the cover has been found to increase from 5% in a city center to 55% in the suburbs [29].

## The effect of patch size on landscape connectivity

Our results, for all dispersal distance thresholds, emphasize the positive correlation between patch size and the importance of the patch for connectivity. In practice, for small distance thresholds (2–20 m), the intracomponent fraction is the most influential for overall values of dIIC. In a situation in which interpatch connectivity is greatly limited, the intrapatch connectivity is crucial and directly related to patch size. For species with large distance dispersal (100–1000 m), direct connectivity with other patches (flux component of dIIC) is decisive. However, since the patch area and flux component are correlated, the largest patches are again more important in maintaining the connectivity within the entire system. Results suggest that the role of patches in serving as stepping stones is rather small, but it has some relevance for species with a moderate dispersal distance (44–100 m).

Individual patch size is important, and it was previously reported that patches of urban green areas need to be larger than 50 ha to prevent a rapid loss of area-sensitive species [94]. It was also previously found in Wrocław city that the area of urban greenery needed to maintain diversified bumblebee (*Apidae*, *Bombini*) communities is at least 30 ha [95]. In our study, the

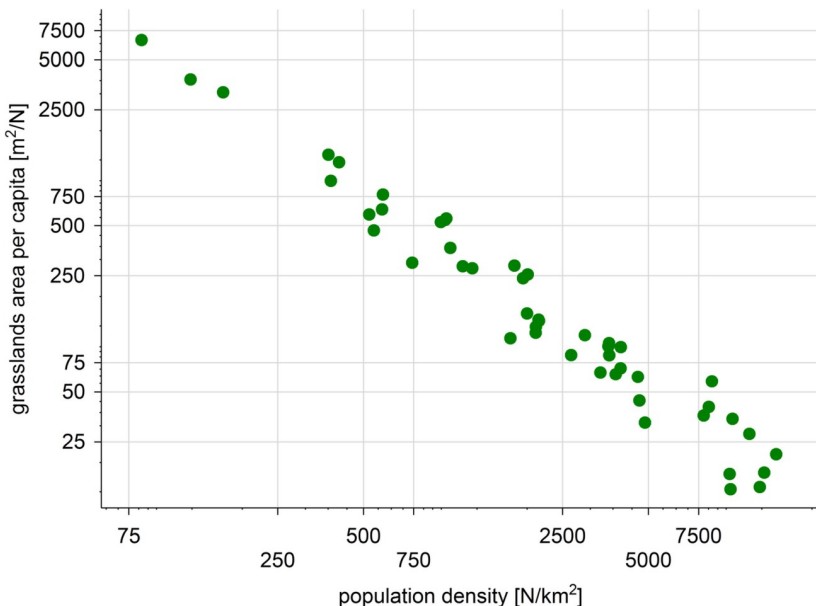

**Fig 6. The relationship between population density and grassland area per capita in Wrocław city.** The values on both axes are on a logarithmic scale.

vast majority of grassland patches were smaller than 50 ha, and they constituted about 50% of all urban grassland area. Thus, it can be assumed that around half of the urban grassland areas in Wrocław city do not efficiently support biodiversity. However, Sehrt [19] showed that even on very small grassland patches (size from 0.02–0.35 ha, average 0.1 ha) with low-intensity management, 6 years after cessation of intense management, the number of vascular plant species increased up to an average of 24 species per patch as a result of spontaneous succession, while only 17 species were recorded on frequently cut lawns. The management changes also caused almost a doubling of species pool from 52 on lawns to 103 on grasslands with low-intensity management [19]. These results suggest that in the case of low-intensity management, urban grasslands with even small patches of habitats are valuable in maintaining the species richness of vascular plants. Investigations of grassland diversity in agricultural areas have yielded similar results. Small grassland remnants as midfield islets and road verges still encompass a substantial part of the grassland species pool, and they may be valuable for reconstructing grassland management at a landscape scale [96, 97]. The small habitat elements increase the total area that is available to grassland species present in the landscape, boosting the spatio-temporal dynamics of grassland communities. They may hence function as a refuge, especially in intensively utilized agricultural landscapes, and they should be regarded as a functional part of a semi-natural grassland network, analogous to a meta-population [98]. In summary, in the case of grasslands, it is important to both protect large habitat patches and maintain an ample amount of habitat in the local landscape around the patches [99].

The results also suggest that the number of components (NC) alone, without relation to their area, can be a somewhat misleading index of landscape structure. In our study, a very limited number of components can make up the majority of an entire habitat area. It is a result of specific, skewed distribution of urban grasslands patch sizes (Fig 2) and seems to be typical for cities [e.g. 39]. Consequently, the number of components that compose a certain percentage (e.g. 50%) of the entire habitat type seems to reflect the structure better than the total number of components (Table 1 and Fig 3).

## Landscape corridors and barriers

The results of a meta-analysis of urban biodiversity variation across different taxonomic groups reveal that, besides patch area, the presence of continuous corridors between patches has the strongest positive effects on biodiversity [94]. This corridor influence is markedly stronger than the distance between patches [94], and it suggests that corridors can be much more effective in promoting urban species richness than stepping-stone habitats. The stepping-stone habitats are viewed as increasing the permeability of a landscape [100], but they simply decrease the distance between patches [101] and do not necessarily provide a functional corridor. In grassland conservation practice in rural landscapes, the linear grassland elements such as road verges and ditches facilitate species persistence and dispersal and the colonization of degraded sites [102]. Moreover, it was found that the connectivity of linear grassland elements is more important to plant species richness than their area for species with short or long distance dispersal [103]. Unfortunately, in spite of the relatively high cover of urban grasslands in Wrocław, their structure is strongly fragmented, with a lack of corridors (Table 1, Fig 3A and 3B). A similar situation was found by Hejkal et al. [39] in Münster, Germany.

Typically, river embankments serve as corridors for migration of plants and animals within cities [104, 105]. The results of our analysis reveal that the presence of a river can enhance dispersal along the embankments as a corridor, but the river itself seems to be a barrier to grassland species migration and divides the city into separate parts (Fig 3A and 3B). Some reports indicate that hydrochory can support seed dispersal of anemo- and zoo-choric tree species [106, 107]. However, in the case of alluvial grasslands in Europe, flooding was not found to increase the dispersal distances of characteristic for this habitat species as *Silaum silaus* and *Serratula tinctoria* [108], and poor dispersal was the main limiting factor for successful restoration of alluvial grasslands [108]. It was also found that flooding does contribute to the density and composition of the seed bank, but most of the imported seeds belong to only a few species. Therefore, flooding is unlikely to substantially enhance the potential species richness of alluvial grasslands [109, 110]. Moreover, genetic analysis has revealed that a large river can be a barrier for seed dispersal [111] and even pollen [112], although islands in rivers can serve as stepping stones for seed dispersal [111].

## Practical implications

The most important strategy for maintaining high levels of urban biodiversity, including grasslands, is to increase the area of habitat patches and create a network of corridors between them [94, 113]. However, the creation of uninterrupted corridors for migration and increase of green areas in practice are only possible in shrinking cities [4, 39], while Wrocław belongs to the group of cities with stable populations [114]. One option for enhancing informal urban green space is to have low-maintenance green tram tracks [115] which can contribute to urban grasslands connectivity. Moreover, other dimensions beyond green areas on the ground need consideration, such as green roofs and walls (vertical gardens), to increase the connectivity [64].

The increase of connectivity of urban grasslands can also rely on increasing their quality, in lieu of their quantity. In a situation of moderate public acceptation of low-intensity grassland management [16], prioritization of patches with the highest importance for connectivity will optimize the selection of patches for improvement. Enhancement of biodiversity through spontaneous succession on lawns released from an intensive management regime seems to be rather restricted because of the extensive fragmentation of urban grasslands. The connection system limits spontaneous migration of species, especially in the city center. Consequently, increasing the species richness of urban grasslands in a city center will require seed addition. To increase the role of spontaneous succession, the use of seed mixtures based on species with

relatively long-range propagule dispersal will be more effective. Unfortunately, most grassland species have a low potential for long-distance dispersal, not exceeding a few meters [108, 116–118]. Dispersal by insects (e.g., ants) and mice is also restricted to several meters [119]. The graph theory provides the tools for selecting the patches that will yield better results for the improvement of connectivity [39]. The dIIC values should be considered, but not as an absolute measure. The spatial context should also be considered to ensure connectivity within the entire city system. Thus, the preferred approach should be patches that have relatively high dIIC value locally (e.g., for central city districts) to ensure provision of ecosystem services for a large number of citizens in city center. The improvement should consist of reducing cutting frequency and applying seed addition, with the seed addition being concentrated on the edges of the patches to increase the probability of species migration to other patches.

It should be stressed that prioritization based on dIIC should be used as a guide to select patches for enhancing their biodiversity, not as tool for determining grassland patches that can be sacrificed. As previously discussed, the value of dIIC is positively corelated with patch size, but small grassland patches can still support a high level of biodiversity [99] and help in connectivity [39].

## Conclusion

Our study reveals that despite a relatively large area of Wrocław being covered by urban grasslands, the connectivity of these grasslands is strongly limited, especially in the city center. The results suggest that when the distribution of habitat patch sizes is skewed, with the smallest patches being dominant, the component number (NC) as a connectivity measure does not reflect the entire landscape structure well. In such a landscape type, only a few components could consist prevailing area of a given habitat. With regard to connectivity, the results on patch prioritization emphasize the importance of the largest patches. However, we argue that even the smallest patches still have value for biodiversity maintenance. Moreover, the prioritization should also consider local demand for urban grasslands, which is much stronger in a highly populated city center than on the peripheries. Study results also highlight the dual role of rivers: the vegetation on embankments can serve as a migration corridor, while the water body itself can be a barrier to migration. Given the impossibility of extending the urban grassland area and creating continuous and structural corridors, increasing connectivity, especially in the city center, should focus on improving grassland quality by seed addition and proper management, as well as developing alternative grassland forms such as green roofs and walls or green tram track lines.

## Supporting information

**S1 Table. Spearman rank correlations between patch area (area), dIIC (dIIC) values, and dIIC components (intra, flux, connector) for distance thresholds (2, 20, 44, 100, and 1000 m).** Values above the diagonal of the matrix are shown. Only significant correlations are presented.
(PDF)

**S1 File. The high-resolution maps of location of urban grassland patches in Wrocław city.** The green color shows the 20% of grasslands patches with the highest dIIC values for a particular dispersal distance; the gray indicates the remaining 80% of patches. The maps are ordered according to increased assumed dispersal distances thresholds: 2m, 20 m, 44 m, 100 m and 1000 m.
(PDF)

**S2 File. Distribution of urban grassland patches against a background of human population density in districts of Wrocław city.** The shapefiles are provided by [66, 69].
(PDF)

**S3 File.**
(DOCX)

## Acknowledgments

The authors would like to thank Oxford Editing (https://oxfordediting.com) for the English language review.

## Author Contributions

**Conceptualization:** Magdalena Szymura, Tomasz H. Szymura.

**Formal analysis:** Hassanali Mollashahi, Tomasz H. Szymura.

**Funding acquisition:** Hassanali Mollashahi, Magdalena Szymura.

**Investigation:** Hassanali Mollashahi, Magdalena Szymura, Tomasz H. Szymura.

**Methodology:** Hassanali Mollashahi, Magdalena Szymura, Tomasz H. Szymura.

**Software:** Hassanali Mollashahi, Magdalena Szymura, Tomasz H. Szymura.

**Supervision:** Magdalena Szymura.

**Writing – original draft:** Hassanali Mollashahi, Magdalena Szymura, Tomasz H. Szymura.

**Writing – review & editing:** Hassanali Mollashahi, Magdalena Szymura, Tomasz H. Szymura.

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
