## [Decision Letter · Decision Letter 0]

13 Oct 2020

PONE-D-20-18120

Connectivity Assessment and Prioritization of Urban Grasslands as a Helpful Tool for Effective Management of Urban Ecosystem Services

PLOS ONE

Dear Dr. Mollashahi,

Thank you for submitting your manuscript to PLOS ONE. After careful consideration, we feel that it has merit but does not fully meet PLOS ONE’s publication criteria as it currently stands. Therefore, we invite you to submit a revised version of the manuscript that addresses the points raised during the review process.

We look forward to receiving your revised manuscript.

Kind regards,

Jun Yang

Academic Editor

PLOS ONE

Journal Requirements:

3. Our internal editors have looked over your manuscript and determined that it is within the scope of our Cities as Complex Systems Call for Papers. This collection of papers is headed by a team of Guest Editors for PLOS ONE: Marta Gonzalez (University of California, Berkeley) and Diego Rybski (Potsdam Institute for Climate Impact Research).

The Collection will encompass a diverse and interdisciplinary set of research articles applying the principles of complex systems and networks to problems in urban science.  Additional information can be found on our announcement page: https://collections.plos.org/s/cities.

If you would like your manuscript to be considered for this collection, please let us know in your cover letter and we will ensure that your paper is treated as if you were responding to this call. If you would prefer to remove your manuscript from collection consideration, please specify this in the cover letter.

4. We note that Figures in your submission contain map/satellite images which may be copyrighted. All PLOS content is published under the Creative Commons Attribution License (CC BY 4.0), which means that the manuscript, images, and Supporting Information files will be freely available online, and any third party is permitted to access, download, copy, distribute, and use these materials in any way, even commercially, with proper attribution. For these reasons, we cannot publish previously copyrighted maps or satellite images created using proprietary data, such as Google software (Google Maps, Street View, and Earth). For more information, see our copyright guidelines: http://journals.plos.org/plosone/s/licenses-and-copyright.

1.     You may seek permission from the original copyright holder of Figure(s) [#] to publish the content specifically under the CC BY 4.0 license.  

Additional Editor Comments (if provided):

Reviewer 1 Major Revision

The authors examined the connectivity of urban grasslands and prioritized the grassland patches with regard to their role in connectivity in an urban landscape. The methodology is sound and the data seems valid. I have some major concerns listed below:

Introduction: It is very poor with limited literature about connectivity assessment, prioritization of urban grasslands and effective management of urban ecosystem services.

Materials and Methods: There is no information about connectivity assessment. Please provide an individual section to introduce connectivity assessment method.

Data analysis: Please separate the data source and method introduction.

Discussion: Some paragraphs are very long. Please pay attention to reasonable segmentation.

The Conclusion section is missing.

There are pictures in the main text and appendices, which seem chaotic.

Some literature to consult:

Local Climate Zone Ventilation and Urban Land Surface Temperatures: Towards a Performance-based and Wind-sensitive Planning Proposal in Megacities. Sustainable Cities and Society, 47(2019):1-11.DOI:10.1016/j.scs.2019.101487.

Assessing the Impacts of Urbanization-Associated Green Space on Urban Land Surface Temperature: A Case Study of Dalian, China. Urban Forestry & Urban Greening. 2017(22), 1–10.doi:10.1016/j.ufug.2017.01.002.

Spatiotemporal variations in greenspace ecosystem service value at urban fringes: A case study on Ganjingzi District in Dalian, China[J]. Science of the Total Environment. 639 (2018) 1453–1461,.doi:10.1016/j.scitotenv.2018.05.253.

Reviewer 2 Major Revision

The authors analyzed the connectivity of urban grasslands and prioritized the grassland patches with regional to their role in connectivity in an urban landscape. The research conclusion is helpful for the effective management of urban ecosystem services. Several visual analysis charts are used to show the conclusion, which is clear and intuitive.

I recommend to publish this paper after addressing the following issues.

(1) It is suggested to add explanation and reference for the calculation principle of connectivity analysis. To prove the scientificity of the analysis results.

(2) This paper mainly discusses the connectivity assessment and prioritization of urban grasslands, but there is less information about prioritization, so it is suggested to add relevant discussions. Secondly, we can discuss the solution of increasing connectivity assessment of urban grasslands.

Reviewers' comments:

Reviewer's Responses to Questions

**Comments to the Author**

1. Is the manuscript technically sound, and do the data support the conclusions?

Reviewer #1: Yes

Reviewer #2: Yes

2. Has the statistical analysis been performed appropriately and rigorously? 

Reviewer #1: Yes

Reviewer #2: Yes

3. Have the authors made all data underlying the findings in their manuscript fully available?

Reviewer #1: Yes

Reviewer #2: Yes

4. Is the manuscript presented in an intelligible fashion and written in standard English?

Reviewer #1: Yes

Reviewer #2: Yes

5. Review Comments to the Author

Reviewer #1: The authors examined the connectivity of urban grasslands and prioritized the grassland patches with regard to their role in connectivity in an urban landscape. The methodology is sound and the data seems valid. I have some major concerns listed below:

Introduction: It is very poor with limited literature about connectivity assessment, prioritization of urban grasslands and effective management of urban ecosystem services.

Materials and Methods: There is no information about connectivity assessment. Please provide an individual section to introduce connectivity assessment method.

Data analysis: Please separate the data source and method introduction.

Discussion: Some paragraphs are very long. Please pay attention to reasonable segmentation.

The Conclusion section is missing.

There are pictures in the main text and appendices, which seem chaotic.

Some literature to consult:

Local Climate Zone Ventilation and Urban Land Surface Temperatures: Towards a Performance-based and Wind-sensitive Planning Proposal in Megacities. Sustainable Cities and Society, 47(2019):1-11.DOI:10.1016/j.scs.2019.101487.

Assessing the Impacts of Urbanization-Associated Green Space on Urban Land Surface Temperature: A Case Study of Dalian, China. Urban Forestry & Urban Greening. 2017(22), 1–10.doi:10.1016/j.ufug.2017.01.002.

Spatiotemporal variations in greenspace ecosystem service value at urban fringes: A case study on Ganjingzi District in Dalian, China[J]. Science of the Total Environment. 639 (2018) 1453–1461,.doi:10.1016/j.scitotenv.2018.05.253.

Reviewer #2: The authors analyzed the connectivity of urban grasslands and prioritized the grassland patches with regional to their role in connectivity in an urban landscape. The research conclusion is helpful for the effective management of urban ecosystem services. Several visual analysis charts are used to show the conclusion, which is clear and intuitive.

I recommend to publish this paper after addressing the following issues.

(1) It is suggested to add explanation and reference for the calculation principle of connectivity analysis. To prove the scientificity of the analysis results.

(2) This paper mainly discusses the connectivity assessment and prioritization of urban grasslands, but there is less information about prioritization, so it is suggested to add relevant discussions. Secondly, we can discuss the solution of increasing connectivity assessment of urban grasslands.

6. PLOS authors have the option to publish the peer review history of their article (what does this mean?). If published, this will include your full peer review and any attached files.

Reviewer #1: No

Reviewer #2: No

---

## [Author Response · Author response to Decision Letter 0]

27 Nov 2020

Academic Editor comments and the author’s responses 

1. The article has been formatted according PLOS One style.

2. The English in the article was checked by professional company, however following your request we checked it again in Oxford Editing Company (https://oxfordediting.com). The small linguistic corrections have been not tracked.

3. We are in touch with Professors Marta Gonzalez and Diego Rybski, and we obtained the information that we should add the message that we are interested in being considered for the collection Cities as Complex Systems in cover letter submitted with improved version of the article.

4. All the maps presented in the article were created by us, using data obtained from Open Data sources. It includes EU open data (the EU-DEM data), Polish Government (the BDOT10k map/database) and Wrocław Spatial Information System (Wrocław district boundaries). The clear statements about the data character and availability is provided by webpages of EU (https://www.eea.europa.eu/data-and-maps/data/eu-dem in the tab “metadada: Rights”) and Marshal Office of the Dolnośląskie Region which represents Polish Government regarding distribution of local geographic data including the BDOT10k set: (http://wgik.dolnyslask.pl/web/start/aktualnosci/-/asset_publisher/3qiOosLOuiIJ/content/baza-danych-obiektow-topograficznych-bdot10k-bez-oplat?redirect=http%3A%2F%2Fwgik.dolnyslask.pl%2Fweb%2Fstart%2Faktualnosci%3Fp_p_id%3D101_INSTANCE_3qiOosLOuiIJ%26p_p_lifecycle%3D0%26p_p_state%3Dnormal%26p_p_mode%3Dview%26p_p_col_id%3Dcolumn-2%26p_p_col_count%3D1). 

In case of Open Data distributed by Wrocław City, the statements about open character of these data is not showed on webpage, thus we attached a correspondence which include the permission from the city authorities (file: ‘City_permission.pdf’). 

In figures captions the references to maps data sources have been added. The acknowledgements for data providers are placed in the ‘Acknowledgement’ part.

5. The ORCID number of the corresponding Author has been added (https://orcid.org/0000-0002-6100-9186).

Additional Editor comments

Reviewer 1 Major Revision

The authors examined the connectivity of urban grasslands and prioritized the grassland patches with regard to their role in connectivity in an urban landscape. The methodology is sound and the data seems valid. I have some major concerns listed below:

Introduction: It is very poor with limited literature about connectivity assessment, prioritization of urban grasslands and effective management of urban ecosystem services.

Answer: We have accomplished the Introduction by additional references devoted to: nature conservation strategies in urbanised areas (Casalegno et al 2017, Pelorosso et al 2017), usage structural connectivity concept for policy makers and managers to plan management strategies of urban green infrastructure (Kong et al 2010, Carlier and Moran 2019, Pauleit et al 2019), and effective management strategies of urban green infrastructure, including ecosystem services related to connectivity (Capotorti et al 2017, Meerow and Newell 2017, Langemeyer et al 2020). To better describe the methodology of connectivity assessment and prioritisation we have added a new section devoted to the methodology in chapter ‘Connectivity analysis and patches prioritisation’ (lines 165-216).

Materials and Methods: There is no information about connectivity assessment. Please provide an individual section to introduce connectivity assessment method.

Answer: We have added an entirely new section describing the methodology of connectivity assessment. We have placed it at the beginning of a section “Connectivity analysis and patches prioritisation” (lines 165-216).

Data analysis: Please separate the data source and method introduction.

Answer: To clarify it we have changed the title of chapter ‘Mapping of grassland patches’ to ‘Data source’. Moreover, we have rearranged the part consisting chapters ‘Data analysis’ and ‘Dispersal distance threshold’: we have merged it in one chapter entitled ‘Methods’ with two sections: ‘Dispersal distance threshold’, and next ‘Connectivity analysis and patches prioritisation’. We hope than now it is clear.

Discussion: Some paragraphs are very long. Please pay attention to reasonable segmentation. The Conclusion section is missing.

Answer: We have changed the entire structure of the Discussion. It includes segmentation it into subsections and shortening of the particular paragraph length. We also have added chapter Conclusions. In the changed discussion we have paid more attention to patch prioritisation, and management options which can increase the connectivity of urban grassland system (section Practical implications).

There are pictures in the main text and appendices, which seem chaotic.

Answer: We have organised figures in the Appendix by providing information regarding their order of presentation, that is according the increasing distance threshold from 2 up to 1000 m. To show it in more clearly way we have merged all the supplementary maps into single Appendix 1. 

The small-size map for distance threshold 44 m is intentionally presented in the main text, as an illustration of conducted analysis (Fig. 5), and additionally showed in the Appendix in large size.

Additionally, we have changed the titles of some figures in text (e.g. Figure 3) to increase the legibility of their order of appearance and content. We have also moved Table 2 and figure 6 from the main text to the Appendix – we hope that it increases the perception of the text. 

Some literature to consult: 

Local Climate Zone Ventilation and Urban Land Surface Temperatures: Towards a Performance-based and Wind-sensitive Planning Proposal in Megacities. Sustainable Cities and Society, 47(2019):1-11.DOI:10.1016/j.scs.2019.101487.

Assessing the Impacts of Urbanization-Associated Green Space on Urban Land Surface Temperature: A Case Study of Dalian, China. Urban Forestry & Urban Greening. 2017(22), 1–10.doi:10.1016/j.ufug.2017.01.002. 

Spatiotemporal variations in greenspace ecosystem service value at urban fringes: A case study on Ganjingzi District in Dalian, China[J]. Science of the Total Environment. 639 (2018) 1453–1461,.doi:10.1016/j.scitotenv.2018.05.253.

Answer: We extend the introduction by an additional paragraph devoted to ecosystem services provided by urban green space. This paragraph is referred from one article suggested by you.

Reviewer 2 Major Revision

The authors analyzed the connectivity of urban grasslands and prioritized the grassland patches with regional to their role in connectivity in an urban landscape. The research conclusion is helpful for the effective management of urban ecosystem services. Several visual analysis charts are used to show the conclusion, which is clear and intuitive.

I recommend to publish this paper after addressing the following issues.

(1) It is suggested to add explanation and reference for the calculation principle of connectivity analysis. To prove the scientificity of the analysis results.

Answer: Following the suggestion we have added an entirely new paragraphs describing the principle of landscape connectivity assessment, basing on graph theory (chapter: Connectivity analysis and patches prioritisation (lines 165-216).

(2) This paper mainly discusses the connectivity assessment and prioritization of urban grasslands, but there is less information about prioritization, so it is suggested to add relevant discussions. Secondly, we can discuss the solution of increasing connectivity assessment of urban grasslands. 

Answer: Following your and the second reviewer suggestion we entirely restructured the discussion part. In the changed discussion, we have paid more attention to habitat prioritization and options for increasing landscape connectivity in urban environment (section Practical implications, (lines 376-409).

---

## [Decision Letter · Decision Letter 1]

10 Dec 2020

Connectivity Assessment and Prioritization of Urban Grasslands as a Helpful Tool for Effective Management of Urban Ecosystem Services

PONE-D-20-18120R1

Dear Dr. Mollashahi,

We’re pleased to inform you that your manuscript has been judged scientifically suitable for publication and will be formally accepted for publication once it meets all outstanding technical requirements.

Kind regards,

Jun Yang

Academic Editor

PLOS ONE

Additional Editor Comments (optional):

Accept

Reviewers' comments:

Reviewer's Responses to Questions

**Comments to the Author**

1. If the authors have adequately addressed your comments raised in a previous round of review and you feel that this manuscript is now acceptable for publication, you may indicate that here to bypass the “Comments to the Author” section, enter your conflict of interest statement in the “Confidential to Editor” section, and submit your "Accept" recommendation.

Reviewer #1: All comments have been addressed

Reviewer #2: All comments have been addressed

2. Is the manuscript technically sound, and do the data support the conclusions?

Reviewer #1: Yes

Reviewer #2: Yes

3. Has the statistical analysis been performed appropriately and rigorously? 

Reviewer #1: Yes

Reviewer #2: Yes

4. Have the authors made all data underlying the findings in their manuscript fully available?

Reviewer #1: Yes

Reviewer #2: Yes

5. Is the manuscript presented in an intelligible fashion and written in standard English?

Reviewer #1: Yes

Reviewer #2: Yes

6. Review Comments to the Author

Reviewer #1: The authors have adequately addressed the comments raised in a previous round of review and this manuscript is now acceptable for publication.

Reviewer #2: (No Response)

7. PLOS authors have the option to publish the peer review history of their article (what does this mean?). If published, this will include your full peer review and any attached files.

Reviewer #1: No

Reviewer #2: No

---

## [Editor Report · Acceptance letter]

14 Dec 2020

PONE-D-20-18120R1 

Connectivity Assessment and Prioritization of Urban Grasslands as a Helpful Tool for Effective Management of Urban Ecosystem Services 

Dear Dr. Mollashahi:

I'm pleased to inform you that your manuscript has been deemed suitable for publication in PLOS ONE. Congratulations! Your manuscript is now with our production department. 

Kind regards, 

on behalf of

Dr. Jun Yang 

Academic Editor

PLOS ONE